# Patients’ Experience of Systemic Treatment of Hepatocellular Carcinoma: A Review of the Impact on Quality of Life

**DOI:** 10.3390/cancers14010179

**Published:** 2021-12-30

**Authors:** Léa Muzellec, Héloïse Bourien, Julien Edeline

**Affiliations:** Department of Medical Oncology, Centre Eugène Marquis, 35000 Rennes, France; h.bourien@rennes.unicancer.fr (H.B.); j.edeline@rennes.unicancer.fr (J.E.)

**Keywords:** hepatocellular carcinoma, quality of life, PROs, treatment management

## Abstract

**Simple Summary:**

Hepatocellular carcinoma remains a serious disease for which curative treatment is only available for about 20% of patients. Because of the severity of the disease and the modest benefit of treatment, quality of life is of paramount importance, especially as its impact on the prognosis of the disease has been demonstrated. Systemic treatments have specific side effects for which close monitoring and symptomatic management improve quality of life.

**Abstract:**

Quality of life (QoL) in oncology is an outcome becoming more and more important and relevant to explore. Some studies have demonstrated its prognostic impact in different cancers, such as colorectal, breast, and prostate cancers, but also in hepatocellular carcinoma (HCC). Different tools have been developed for assessing quality of life, some general, such as EORTC QLQ-C30, but also specific tools depending on cancer origin which seem to be more pertinent for patients. Systemic treatments and specific symptoms due to cancer evolution could decrease quality of life. For approval of new systemic treatments, authorities ask for benefit in terms of efficacy but also benefit in quality of life, which is crucial for patients. This review reports data about QoL in HCC, including specific tools used, impact of systemic treatments and prognosis for QoL for HCC patients. Management of adverse events is essential to enhance compliance with treatment and quality of life. Assessing quality of life in clinical trials appears quite systematic, but its application in clinical routine requires development.

## 1. Introduction

Hepatocellular carcinoma (HCC) is the most common liver primary tumor with a high rate of mortality. Only about 20% of patients will access curative options, and even in these cases relapse rates are high. Most patients are diagnosed at an advanced stage and are not suitable for curative treatments. Therapeutic options are locoregional treatments such as resection, chemo-embolization, radio-embolization or radiotherapy, and systemic treatments, such as anti-angiogenics, or more recently, immunotherapy. Given the severity of this disease and the modest benefit of treatments, quality of life (QoL) of the patient becomes very relevant and should be a major concern to clinicians.

During the last decade, sorafenib, lenvatinib, atezolizumab-bevacizumab combination, cabozantinib, regorafenib and ramucirumab have proven efficacy with longer PFS or OS as first or second lines in randomized trials. These treatments have potential side effects that could impact QoL which has been shown to be a clinical outcome as important as PFS or OS in the management of patients monitored for hepatocellular carcinoma.

This review aims to summarize current evidence about QoL in HCC, as well as the impact of systemic treatment on QoL in advanced cancer and specifically in HCC.

## 2. Methods

This review was written as a narrative review. The objective was to summarize evidence regarding QoL in HCC, as well as illustrating pertinent information relating to systemic treatment of other solid tumors, that could be applied to HCC treatment. Literature research was made via Pubmed in July 2021 using the terms “hepatocellular carcinoma”, “solid tumors”, “quality of life”, “targeted therapies”, “immune checkpoint inhibitors”, “immunotherapy” and “adherence”. Additional references were retrieved from articles. As this was not a systematic review, no formal inclusion/exclusion criteria were selected. However, we cited studies that provided information regarding evaluation of quality of life under systemic treatment, focusing on targeted therapies and immunotherapies, in HCC and other solid tumors. We also selected studies referring to the follow-up of QoL during systemic treatment of solid tumors, and studies focusing on the management of toxicities with the goal of improving patients’ QoL.

## 3. Results

### 3.1. Quality of Life and HCC

#### 3.1.1. Main Tools for Measuring Quality of Life in Patients Followed for HCC

Health-related QoL has been defined as physical, mental and social well-being. Several scales have been validated to measure the specific QoL of liver cancer patients (Table 1).

FACT–Hep (Functional Assessment of Cancer Therapy-Hepatobiliary) [1] is a 45-item quality of life scale, containing 18 items specific to hepatobiliary disease. The minimally clinically important difference (MCID) of the FACT-Hep score is 8 to 9 points. EORTC QLQ-HCC18 (Quality of Life Questionnaire) [2] includes 18 specific items and completes the QLQ-C30, a health-related QoL score for general cancer. For the different scales of the EORTC QLQ-C30 tool, a 5 to 10-point change from baseline (either deterioration or improvement) is ‘a little’ change, a 10 to 20-point change is a ‘moderate’ change, and a ‘very large’ change corresponds to a change greater than 20. FHSI (FACT Hepatobiliary Symptom Index) [3] uses 8 items from FACT-Hep and is intended to evaluate symptoms (e.g., pain, fatigue, nausea, weight loss, and jaundice). QOL-LC (Quality of Life-liver cancer) [4] included symptoms and adverse events in a Chinese cohort with liver cancer. The MCID of FHSI and QOL-LC have not been defined. When comparing different treatment modalities, the QoL adjusted life-years (QALY) measure, evaluating the survival with preserved QoL, is frequently used to quantify the benefit of an intervention.

#### 3.1.2. Description of Quality of Life of HCC Patients

A review of 36 articles analyzed the impact of disease and treatment on quality of life using EORTC QLQ-C30, EORTC QLQ-HCC18, FACT-Hep and FACT-G [4]. Compared to patients with chronic liver disease, patients with HCC had worse physical well-being and compared to the general population, worse QoL on physical, (*p* < 0.001), psychological (*p* < 0.001) and functional (*p* < 0.001) items, but better QoL on family and social level (*p* < 0.001). QoL was better after hepatic surgery, but not after hepatic intra-arterial therapy and radiotherapy.

The study by Steel et al. included in Fan’s meta-analysis included 272 patients (83 HCC, 51 chronic liver disease, 134 general population). The objective was to compare the FACT-Hep score in patients followed for HCC vs. chronic liver disease vs. the general population. Patients followed for chronic liver disease or HCC had, respectively, better social and family quality of life compared to the general population. Patients followed for HCC had a lower overall quality of life than those followed for chronic liver disease (*p* = 0.032) [5].

Another trial included in Fan’s meta-analysis (Kondo et al.) compared QoL in 97 patients with HCC and 97 patients with chronic liver disease. The latter was similar between the 2 groups but lower than in the general population. The liver function, and in particular the albumin level, were strongly predictive of QoL in multivariate analysis [6].

QoL improves with better liver function (as characterized by Child-Pugh score, albumin and bilirubinemia), localized stage of disease and absence of recurrence, and decreases with pain, fatigue, nausea and performance status.

Liver function is correlated with quality of life. Albumin, bilirubin, alkalin phosphatase (ALP), albumin to platelet ratio, albumin to ALP ratio, Child-Pugh classification, albumin-bilirubin (ALBI) grade, model for end-stage liver disease (MELD) and ascites have been shown as independent prognostic factors for QoL [7].

Demographically, QoL decreases with depressive symptoms, female gender, and feelings of uncertainty, and increases with age and satisfaction with medical services.

Several trials have shown a positive impact of psychosocial interventions on reducing negative feelings and strengthening QoL.

#### 3.1.3. Prognostic Impact of QoL in HCC Patients

EORTC QLQ-HCC18 has been found to be prognostic for overall survival, independent of HCC stage and liver function [8]. Other trials have also demonstrated the prognostic value of QoL for unresectable HCC, regardless of the etiology of cirrhosis: HBV with the EORTC QLQ-C30 score [9] or alcoholism with the Spitzer QoL Index, a cancer-specific QoL measurement [10]. Diouf et al. also confirmed in patients of the CHOC cohort the prognostic value of quality of life measured by the EORTC QLQ C30 questionnaire [11,12].

In conclusion, HCC and altered liver function (either pre-existing or due to HCC) have a significant impact on patients’ quality of life and QoL may be an important prognostic marker.

### 3.2. Tolerance and Compliance of Treatments for HCC and Other Indications

#### 3.2.1. Tolerance of Systemic Treatments (TKI and Immunotherapy) and the Impact on QoL

Toxicity of TKIs in HCC

The rate of discontinuation due to toxicity was similar for sorafenib (11%), regorafenib (10%), lenvatinib (9%) and cabozantinib (16%) [13,14,15,16,17]. The incidence of toxicities greater than or equal to grade III was 45% to 75%, with mainly: fatigue, diarrhea, hand-foot skin reaction, nausea, vomiting, anorexia, hypertension, and weight loss. The patterns of toxicities were very similar between sorafenib, regorafenib and cabozantinib, but lenvatinib showed a slightly different pattern, with less hand-foot skin reaction, but more hypertension.

Toxicity of immunotherapy for all indications

Anti-PD-1, anti-PD-L1, and anti-CTLA-4 have adverse events (AEs) that are different from TKIs, which may be infusion-related or immunologically based. These AEs can occur up to several months after discontinuation of treatment. With combination immunotherapy involving anti-CTLA-4, tolerability is generally more difficult, with nearly 95% of patients reporting at least one AE, 55% of which are grade 3–4.

The most frequently reported AEs are skin toxicity (45% with ipilimumab, and 35% with nivolumab and pembrolizumab), asthenia (with anti-PD1: 16–37% and anti-PDL1: 12–24%) and endocrinopathies (e.g., dysthyroidism, with incidence of 5–10%, hypophysitis, diabetes). Multiple toxicities have also been observed including hepatic, gastrointestinal, pulmonary, neurological, cardiac, rheumatological, renal, ocular and hematological [18]. Grade 3 to 4 AEs from anti-CTLA-4 are more frequent than for anti PD-1 or PDL-1 [18]. For metastatic melanoma patients, 10 to 27% treated with ipilimumab [19] versus 12 to 20% with nivolumab [20,21] develop grade 3–4 AEs. In advanced unresectable stage III or IV melanoma, patients treated with ipilimumab had a lower QoL than those treated with pembrolizumab [22].

Toxicity of immunotherapy in HCC

There does not appear to be a different safety profile for patients treated with immunotherapy for HCC than for those with immunotherapy for other indications [23].

In conclusion, the adverse effects of systemic therapies in HCC put patients at risk for QoL impairment.

#### 3.2.2. Quality of Life and QALYs under TKI and Immunotherapy

The assessment of quality of life in studies is, in most cases, a secondary endpoint.

QoL of patients undergoing TKI for diverse cancers

Epidermal growth factor inhibitors (EGFRI) are frequently prescribed for patients with solid tumors. However, most patients treated with EGFRI would present dermatologic toxicities such as pruritus, xerosis, papulopustular eruption, paronychia, mucositis or hair changes. These AEs would usually appear during the first month of treatment. Although the presence and the severity of cutaneous symptoms has a positive correlation with response to treatment [24,25], they also have a negative impact on QoL. In a sub-analysis of 85 patients treated by EGFRI, xerosis and pruritus had the most negative impact on QoL. In this study, QoL was evaluated during the first six weeks of treatment of EGFRI, using five different questionnaires: the DERETT-P (Dermatological Reactions Targeted Therapy–Patients), the Functional Assessment of Cancer Therapy-EGFRI (FACT-EGFRI-18), the Functional Assessment of Cancer Therapy-General (FACT-G), the 36-Item Short Form Health Survey (SF-36), and the Skindex-16 [26]. Despite these dermatologic toxicities impacting QoL, QoL seems to be better for patients treated with EGFRI than for patients treated by chemotherapy; in a cohort of 345 patients with lung adenocarcinoma EGFR mutated, patients treated with afatinib had a better QoL compared to patients treated with chemotherapy [27].

Treatment with a combination of two targeted therapies is not necessarily associated with an increase of toxicities and negative impact on QoL. For example, in the COMBI-v [28] randomized study, patients treated with dabrafenib and trametinib had a better QoL compared to patients treated with vemurafenib. QoL was assessed with three questionnaires: European Organisation for Research and Treatment of Cancer Quality of Life (EORTC QLQ-C30), EuroQoL-5D (EQ-5D), and Melanoma Subscale of the Functional Assessment of Cancer Therapy-Melanoma (FACT-M)).

QoL of patients undergoing immunotherapy for diverse cancers

Immunotherapy has a better safety profile than chemotherapy or targeted therapies, and frequently improves QoL as compared with other treatment.

Among patients with metastatic melanoma, advanced squamous-cell non-small-cell lung cancer (NSCLC) or with recurrent or metastatic squamous cell carcinoma of the head and neck, QoL was significantly better with anti-PD-1 than with chemotherapy [29,30,31].

In the phase 3 CheckMate-214 trial [32], patients with an advanced or metastatic renal clear-cell carcinoma treated with nivolumab plus ipilimumab had a significantly better QoL than patients treating with sunitinib. QoL was evaluated with three questionnaires: one for the general population, EQ-5D-3L (EuroQol five-dimensional three level), and two cancer-specific questionnaires, the FACT-G (Functional Assessment of Cancer Therapy-General) and the FKSI-19 (Functional Assessment of Cancer Therapy Kidney Symptom Index-19).

The same results were observed with everolimus, another TKI, compared to nivolumab in the phase 3 CheckMate 025 trial, in a cohort of patients with advanced renal cell carcinoma in 2nd or 3rd line treatment [33].

QoL during systemic therapy for patients followed for HCC

Sorafenib was the first TKI approved in advanced HCC. The median time to symptomatic progression (which was defined as either a decrease of four or more points from the baseline score on the FHSI8 questionnaire or an ECOG status of four or death) was not significantly longer in the sorafenib group, compared to placebo group [14].

Other TKIs have been approved for advanced HCC. In first-line treatment, lenvatinib was not inferior to sorafenib, with a median survival time of 13.6 months compared to 12.3 months with sorafenib (hazard ratio 0.92, 95% CI 0.79–1.06) [16]. Patients treated with lenvatinib or sorafenib experienced a deterioration of QoL based on EORTC QLQ-C30 and EORTC QLQ-HCC18 questionnaires [16]. There was no difference in the summary score between the two groups, but for three items from QLQ-C30 (role functioning, pain and diarrhea) and two items from QLQ-HCC18 (nutrition and body image) deterioration was earlier in the sorafenib group than in the lenvatinib group. Kobayashi et al [34] had evaluated the cost-effectiveness of lenvatinib compared with sorafenib for Japanese HCC patients included in the REFLECT trial; due to its better PFS, lenvatinib had improved QALYs. In their analysis, the incremental LYs and QALYs were 0.27 and 0.23, respectively, for lenvatinib compared with sorafenib, and the simulation showed that lenvatinib was preferred in 81.3% of the scenarios when the payer’s willingness to pay per QALY was 5.0 million Japanese Yen (corresponding to 44,000 US dollars).

In second-line treatment, three antiangiogenics have been approved: regorafenib, ramucirumab and cabozantinib. Regorafenib and ramucirumab [35] did not improve QoL compared to placebo. Regorafenib was associated with significantly lower QoL than placebo (using FACT-Hep), but the difference did not reach the clinically-significant threshold [15]. Cabozantinib increased QALYs significantly but modestly compared to placebo (+0.092, 95% CI 0.016 to 0.169; *p* = 0.018), in a post hoc analysis [36] of CELESTIAL trial [17].

Regarding immunotherapy, in patients with an advanced HCC previously treated with sorafenib, pembrolizumab did not deteriorate QoL vs. placebo [37]. Compared to sorafenib, patients treated with nivolumab [38] and an atezolizumab-bevacizumab combination [39] had a significantly better quality of life. Under atezolizumab-bevacizumab, the median time to deterioration of QoL, according to the EORTC QLQ-C30, was 11.2 months (95% CI: 6.0-not evaluable) vs. 3.6 months (95% CI: 3.0–7.0) for sorafenib.

In conclusion, most TKIs have not shown any benefit in terms of improvement of QoL. Some may have been associated with a negative effect. Lenvatinib appears to have a slightly better patient experience profile than sorafenib. Anti-PD-1 and atezolizumab-bevacizumab appear to be associated with better QoL than sorafenib.

#### 3.2.3. Adherence to Treatment

With the approval of more and more oral anticancer treatments, patient adherence becomes an important issue. Studies have shown that better adherence is associated with improved efficacy [40] and with a reduction in healthcare costs (i.e., in reducing rate of hospitalization and its duration and physician visits) [41]. There is no gold standard for measurement of adherence to an oral anti-cancer therapy. Some authors have defined adherence as when the patient takes more than 80% of the pills prescribed [42]. Analysis of real-world data are important to assess treatment compliance.

TKI Adherence in Diverse Cancers:Advanced renal cell carcinoma (RCC):

In a cohort of patients with an advanced RCC, 81% of patients had a compliance rate ≥ 80% with sorafenib, sunitinib and everolimus, but also, which limits the analysis, with intravenous therapies, such as bevacizumab, temsirolimus or interferon [43].

High compliance and persistence of pazopanib has been measured in the real-world setting [44]. Pazopanib persistence was measured using the percentage of patients remaining on therapy at 30, 60 and 90 days, time to discontinuation; and proportion of days covered by the treatment (PDC, the ratio of the number of days covered by a patient’s deliveries to the number of days to be covered by the treatment). Compliance was measured by medication possession ratio (MPR), the ratio of the number of days of treatment dispensed to the number of days the patient should be in possession of the drug.

Metastatic colorectal cancer (mCCR):

For refractory metastatic colorectal cancer, both chemotherapy with trifluridine/tipiracil and TKI with regorafenib have been approved. Both treatments showed significant improvement in median overall survival [45,46]. In a retrospective cohort of mCCR, 1630 patients were treated with trifluridine/tipiracil and 1425 patients were treated with regorafenib [47]. Adherence was evaluated using MPR and PDC at 3 months. The mean MPR and the mean PDC at 3 months were significantly greater in the trifluridine/tipiracil group than for the regorafenib group (mean MPR, 0.91 vs. 0.87, *p* < 0.001 and mean PDC at 3 months, 0.71 vs. 0.59, *p* < 0.001). A Japanese retrospective study explored clinical factors that may affect adherence to regorafenib [48]. The main factors identified for non-adherence to regorafenib were hand-foot-skin reaction, fever, rash and pain.

Breast cancer:

In a retrospective analysis, adherence to everolimus treatment was evaluated for patients treated for hormone-sensitive, HER2-negative, metastatic breast cancer patients [49]. Adherence was evaluated using MPR during second-, third- or fourth-line therapy. Across the 645 patients, the median MPR ranged from 0.9 to 0.92.

Non-small cell lung cancer (NSCLC):

In a prospective study, compliance of erlotinib was evaluated for 62 patients during the first four months of treatment [50]. The mean PDC was 96.8 ± 4.0%. Factors associated with lower adherence were older age, ocular symptoms and stomatitis.

In these studies, the median duration of treatment was short (about a few months) and the literature was poor for data on long-term compliance of TKI. One explanation could be the use of these therapeutics in advanced cancer settings.

In conclusion, in tumors where treatments are given on a prolonged basis, and where multiple lines of treatment exist, compliance with TKIs appears to be maintainable over the long term. The main factors limiting compliance appear to be side effects.

Compliance of TKI for patients with HCC

In the phase III trial evaluating sorafenib in first line treatment for patients with HCC, 76% of patients in the sorafenib group received >80% of the planned daily dose [14]. Analysis of the adjuvant STORM trial [51] indicated difficulty in maintaining treatment with AEs as the most frequent reason for discontinuation of treatment in the sorafenib arm (133, 24%) versus in the placebo arm (41, 7%).

No compliance data are available for trials with lenvatinib or cabozantinib.

The RESORCE phase III trial [15], evaluating 2nd line regorafenib versus placebo, showed better tumor control with regorafenib (65 vs. 36% *p* < 0.0001), but more than half of the patients (54 vs. 10%) suspended treatment or had the dose reduced.

In conclusion, for patients treated for hepatocellular carcinoma, compliance is imperfect with TKIs. The adjuvant setting also seems less conducive to treatment compliance.

#### 3.2.4. Symptomatic Management Improves Quality of Life

Each pharmaceutical class has specific AEs that can alter patients’ QoL. Symptomatic management and preventive care for these AEs are necessary. For example, preventive application of urea-based cream under sorafenib, could significantly delay the median time to first occurrence of hand-foot-skin reactions, and improved QoL [52].

In order to enhance compliance and tolerance of oral cancer therapies, some teams are developing educational programs with the involvement of nurses with a significant benefit of maintaining an optimal therapeutic dose with early toxicity management [53,54,55,56,57].

In conclusion, better management of adverse events appears to improve QoL.

#### 3.2.5. Continuous Evaluation of Patient Reported Outcomes (PRO)

Symptomatic management of AEs leads to better quality of life. However, there is a gap between symptoms reported by patients (PRO) and those evaluated by clinicians [58,59]. Systematic measurement of patient symptoms, using patient reported outcomes (PRO), can prevent emergency visits or hospitalization [60], ameliorate communication between clinician and patients [61], and improve overall survival [62], symptoms and quality of life [63]. To help clinicians access exhaustive monitoring of PRO in routine settings, web applications have been developed [64,65]. The connection from the patient’s home to the physician is immediate and allows medical action leading to improved clinical outcomes.

Evaluation of PRO is commonly used in clinical trials and its application in clinical practice is growing. Monitoring PRO routinely is not only feasible but also has benefits in terms of QoL (Figure 1).

## 4. Conclusions

HCC remains a serious disease with modest treatment efficacy. Quality of life is, therefore, an essential element to consider in the management of patients, especially since its impact on overall survival has been demonstrated. Adverse effects of HCC treatments must be prevented and followed up in order to limit the deterioration of quality of life and improve patients’ adherence to treatment. Continuous evaluation of PROs might help to better assess patients’ experience under treatment, and eventually also lead to improved oncological outcomes.

## Figures and Tables

**Figure 1 cancers-14-00179-f001:**
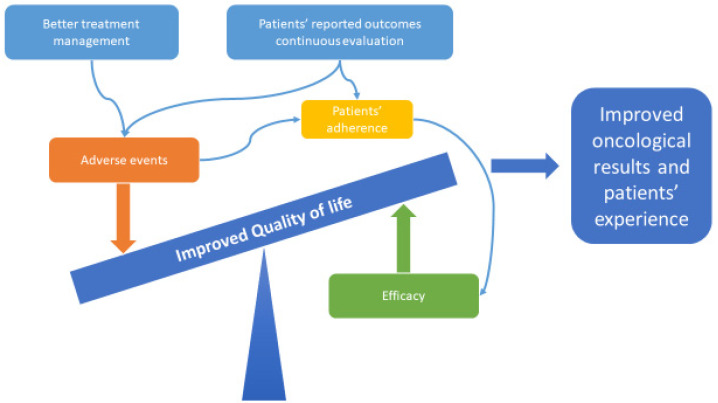
Treatment and quality of life.

**Table 1 cancers-14-00179-t001:** QoL scales and symptoms assessed.

EORTC QLQ-C30	EORTC QLQ-HCC18	FACT-Hep	FSHI	QOL-LC
Global health status/quality of life Physical functioning Role functioning Cognitive functioning Social functioning Fatigue Nausea/Vomiting Pain Dyspnea Sleep disturbances Appetite loss Constipation Diarrhoea Financial difficulties	Shoulder pain Abdominal pain Wegth Eating Fever Jaundice Thirst Abdominal swelling Figure of the abdomen Vitality Pruritus Taste Temperature regulation Muscle loss Nutrition Sleepiness Sexuality	Pain Weight loss Digestion Diarrhea Fever Fatigue Jaundice Dry mouth Stomach pain Change in appearance Daily activities Pruritus Taste Chills Control of bowels Constipation Appetite	Pain Pain Weight loss Fatigue Jaundice Nausea	Pain Weight loss Digestive problem Diarrhea Fever

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
