# Peer review of "Patients’ Experience of Systemic Treatment of Hepatocellular Carcinoma: A Review of the Impact on Quality of Life"

_cancers, 2021, doi:10.3390/cancers14010179_

Round 1

Reviewer 1 Report

The aim of this review was to summarise current evidence about QoL in HCC, as well as the impact of systemic treatment on QoL in advanced cancer and specifically in HCC. This is an important topic area, although it is unclear why the question about the impact of treatment on QoL was broadened to different types of cancer, rather than just HCC, which was the primary topic of the review (due to a lack of information specific to HCC or to compare the impact of treatment on QoL between HCC and other cancers?).

This does not appear to be a systematic review; there are no review methods reported. Therefore, it is not possible to assess the appropriateness and quality of the included studies, whether the results were a complete and accurate summary of all of the available data, etc. Whilst the review does not claim to be systematic, this makes it difficult to complete the ratings about whether the work is a significant contribution to the field, whether the work was comprehensively described (there was no methods section), whether the work was scientifically sound (again, difficult to assess with no methods section) and whether references were appropriate. The manuscript presents interesting information on quality of life of HCC patients and the association between quality of life and treatment toxicity. It is generally well written, although there are some minor corrections to spelling/grammar required. However, as it is not a systematic review it is not possible to determine the validity of the results and appropriateness of the conclusions.

Author Response

We thank the reviewer for this important comment. Indeed, this is a narrative review rather than a systematic review. We added a “method section”.

Reviewer 2 Report

This paper conducted a review to summarize current evidence about quality of life (QoL) in hepatocellular carcinoma (HCC), as well as the impact of systemic treatment on QoL in advanced cancer and specifically in HCC. I do have some comments as listed below in the order noted.

Comment 1: The authors mentioned that a review of 36 articles analyzed the impact of disease and treatment on QoL. The quality of the study is very important, especially in a retrospective review-based study. For this reason, please clarify the inclusion criteria and exclusion criteria of study collection in the Methods section and please provide a flowchart immediately at the subsection of Study Collection.

Comment 2: Although the EORTC QLQ-C30, EORTC QLQ-HCC18, FACT–Hep and FACT–G instruments allow measurement of treatment-related changes in Patients’ Experience of Systemic Treatment, a key question is whether these changes are deemed to be of value by the patient. A simple comparison of numbers, even if demonstrating statistical significance, does not per se imply that the difference in QoL reached a level of benefit perceptible to the patient. Interpreting Patients’ Experience of Systemic Treatment must therefore consider the concept of minimal clinically important difference (MCID). Please determine and summarize the MCID and associated factors of each of the four measures in the subsection of Study Outcomes.

Comment 3: Please define and provide the details related to QALYs in the subsection of Quality of life and QALYs under TKI and immunotherapy.

Comment 4: Please conduct the Method section in the present study.  

Author Response

This paper conducted a review to summarize current evidence about quality of life (QoL) in hepatocellular carcinoma (HCC), as well as the impact of systemic treatment on QoL in advanced cancer and specifically in HCC. I do have some comments as listed below in the order noted.

Comment 1: The authors mentioned that a review of 36 articles analyzed the impact of disease and treatment on QoL. The quality of the study is very important, especially in a retrospective review-based study. For this reason, please clarify the inclusion criteria and exclusion criteria of study collection in the Methods section and please provide a flowchart immediately at the subsection of Study Collection.

We thank the reviewer for this important comment. We modified the methods section accordingly.

Comment 2: Although the EORTC QLQ-C30, EORTC QLQ-HCC18, FACT–Hep and FACT–G instruments allow measurement of treatment-related changes in Patients’ Experience of Systemic Treatment, a key question is whether these changes are deemed to be of value by the patient. A simple comparison of numbers, even if demonstrating statistical significance, does not per se imply that the difference in QoL reached a level of benefit perceptible to the patient. Interpreting Patients’ Experience of Systemic Treatment must therefore consider the concept of minimal clinically important difference (MCID). Please determine and summarize the MCID and associated factors of each of the four measures in the subsection of Study Outcomes.

We thank the reviewer for this important comment. We added the MCID in the 3.1.1 section.

Comment 3: Please define and provide the details related to QALYs in the subsection of Quality of life and QALYs under TKI and immunotherapy.

We thank the reviewer for this comment. We added the figures in this section

Comment 4: Please conduct the Method section in the present study.  

We thank the reviewer for this important comment. We added a “method section”.

Round 2

Reviewer 2 Report

The authors have replied to the reviewer's comments. I do have one comment as listed below.

Comment 1: For further validation of the significant association observed between systemic treatment and QoL for HCC, please provide a Table to summarize current evidence about the impact of systemic treatment on QoL in advanced cancer and specifically in HCC.

Author Response

Thank you for your comment. Please find attached the table. 

We confirm this article is a review.
